# Multidirectional desymmetrization of pluripotent building block en route to diastereoselective synthesis of complex nature-inspired scaffolds

Vunnam Srinivasulu[1], Paul Schilf[2], Saleh Ibrahim[2], Monther A. Khanfar[3], Scott McN Sieburth[4], Hany Omar [1,5,6], Anusha Sebastian[1], Raed A. AlQawasmeh[3], Matthew John O'Connor[7] & Taleb H. Al-Tel [1,5]

Octahydroindolo[2,3-a]quinolizine ring system forms the basic framework comprised of more than 2000 distinct family members of natural products. Despite the potential applications of this privileged substructure in drug discovery, efficient, atom-economic and modular strategies for its assembly, is underdeveloped. Here we show a one-step build/couple/pair strategy that uniquely allows access to diverse octahydroindolo[2,3-a]quinolizine scaffolds with more than three contiguous chiral centers and broad distribution of molecular shapes via desymmetrization of the oxidative-dearomatization products of phenols. The cascade demonstrates excellent diastereoselectivity, and the enantioselectivity exceeded 99% when amino acids are used as chiral reagents. Furthermore, two diastereoselective reactions for the synthesis of oxocanes and piperazinones, is reported. Phenotypic screening of the octahydroindolo[2,3-a]quinolizine library identifies small molecule probes that selectively suppress mitochondrial membrane potential, ATP contents and elevate the ROS contents in hepatoma cells (Hepa1–6) without altering the immunological activation or reprogramming of T- and B-cells, a promising approach to cancer therapy.

[1] Sharjah Institute for Medical Research, University of Sharjah, P.O. Box 27272 Sharjah, UAE. [2] Lübeck Institute of Experimental Dermatology, University of Lübeck, Ratzeburger Allee 160 23538, Lübeck, Germany. [3] Department of Chemistry, University of Jordan, 11942 Amman, Jordan. [4] Department of Chemistry, Temple University, 201 Beury Hall, Philadelphia, PA 19122, USA. [5] College of Pharmacy, University of Sharjah, P.O. Box 27272 Sharjah, UAE. [6] Faculty of Pharmacy, Beni-Suef University, Beni-Suef 62514, Egypt. [7] New York University, Abu Dhabi, P.O. Box 129188 Saadiyat Island, Abu Dhabi, UAE. Correspondence and requests for materials should be addressed to S.I. (email: Saleh.lbrahim@uksh.de) or to T.H.A.-T. (email: taltal@sharjah.ac.ae)

**N**atural products encompass a wealth of structural diversity within defined, nonplanar scaffolds; however, there are well-documented hurdles to their use in screening campaigns because of their limited availability relative to the quantities needed for structure activity relationship (SAR) development and clinical trials[1–3]. Thus, the recent decade has witnessed an upsurge in the development of privileged substructure diversity-oriented synthesis (DOS) strategies for the de novo construction of nature-inspired compounds needed for phenotypic-screening campaigns[4–6]. One such strategy represents the use of a single pluripotent functional group that can be decorated through reactions with variety of reagents, thereby empowering the synthesis of skeletally diverse compound collections with high 3D-content[7–9]. There is a growing consensus that greater 3D-content and numbers of stereocenters within a specific library will enhance the selectivity and potency toward a given target, hence increasing the hit to lead success rate across several targets from a single library[9,10]. In addition, it has been recently reported that decreasing the aromatic ring count in a compound library correlates favorably with decreasing clinical toxicity and compound attrition rate in clinical trials[11,12]. These investigations have spurred a growing belief in the advantage of increasing the percentage of sp$^3$-hybridized atoms within a compound collection used for phenotypic screening.

In this context, one of the important classes of natural products is the octahydroindolo[2,3-a]quinolizine monoterpene indole alkaloids, comprised of more than 2000 members and among the most studied natural product classes owing to their diverse biological activities and synthetic potential associated with these scaffolds[13–16]. This molecular framework is produced by an array of plants and microorganisms. Several members of this monoterpene family possess biological activities useful for the treatment of many disease states (Fig. 1)[13–16]. Intriguingly, the crosstalk of these natural products with their complementary biological targets is tightly linked to both the asymmetry of ring-fusion patterns and arrangement of peripheral arms surrounding a common 6–5–6–6 core (Fig. 1)[15–18].

The structures in this class vary greatly in the substitution about the indole moiety (Fig. 1), including several members that are epimeric at C(3), possess variable ring-fusion patterns and

display interesting and contrasting biological activities[15,16]. Thus, the development of a stereo-controlled, step-economic, and atom-economic strategy for their access would be a remarkable achievement[18,19]. Such an approach would facilitate comprehensive phenotypic-screening studies that might lead to the discovery of chemical probes for multiple phenotypes. Taken together, a significant amount of attention has been devoted to their access, especially the archetypal member, reserpine[20]. Among others, Sarpong and Stork reported innovative approaches for the synthesis of various scaffolds of this class of natural products[18,21]. You[17] and Zhai[22] described an elegant synthesis of these scaffolds employing the Pictet–Spengler reaction. Additional important contributions to the synthesis of this class of natural products were also reported by Hamada[23], Poupon[24], and Amat[25], however a major limitation of existing strategies is the use of protracted, multistep syntheses to prepare the requisite building blocks. In addition, many reported methods require transition-metal catalysis and harsh conditions to promote the desired chemistry. Despite the importance of these stepwise methods, a general and modular strategy for the preparation of various analogues of the octahydroindolo[2,3-a]quinolizine monoterpene indole alkaloids family is highly desirable. With an eye toward exploiting the potential biological significance of a library of this scaffold, we set out to develop a general route to this molecular architecture that would facilitate straightforward diversification of functional groups surrounding the 6–5–6–6 nucleus.

Borrowing inspiration from these findings, we aim to assemble a pilot library of natural product-inspired analogues and isosteres thereof that might elicit the desired interactions with a range of therapeutic targets. Thus, here we describe a one-step chemo-, diastereo- and enantio-selective protocol for the construction of densely functionalized octahydroindolo[2,3-a]quinolizine systems with three to five contiguous chiral centers. This protocol employs tandem cycloaddition processes utilizing Pictet-Spengler/aza-Michael addition, the products of which constitute the basic framework of the biologically significant natural products yohimbine, venenatine, alstovenine and tangutorine (Fig. 1)[15,18]. In another aspect, the discovery of oxocanes via aza-Michael addition/Mannich cascade using aniline derivatives as one of the

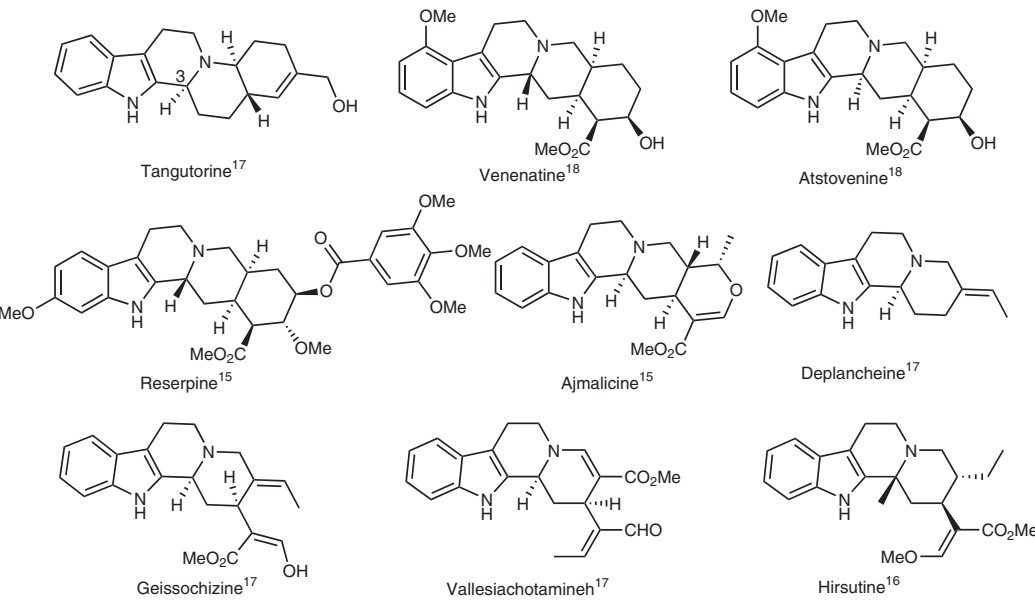

**Fig. 1** Representative examples of bioactive octahydroindolo[2,3-a]quinolizine systems

reaction components, is described. Furthermore, a third seren-dipitous cascade to access polysubstituted piperazinones in a 100% enantioselective process, is reported. Claisen rearrangement forms a key step of the latter cascade which proceeds through reacting amino acids with the same pluripotent building block. Additionally, post-pairing differentiation of the octahydroindolo [2,3-a]quinolizine products, using osmium tetroxide ($OsO_4$) and N-bromosuccinimide (NBS) as reaction promoters, allows for the discovery of rearrangement reactions that enrich the library with yet more complex molecular shapes. These findings illustrate a concept of triply divergent, diversity-oriented syntheses as a tool to access skeletally diverse collections of high 3D-content compounds.

The cellular monitoring of adenosine triphosphate (ATP) production in hepatoma cells (Hepa1–6), the octahydroindolo [2,3-a]quinolizine pilot library yields a chemotype that effectively blocks ATP synthesis, mitochondrial membrane potential ($\Delta\Psi m$), elevates the level of reactive oxygen species (ROS) and suppresses cellular proliferation without altering the immunological activation or the metabolic reprogramming of T- and B-cells, representing a promising approach for cancer therapy.

## Results

**Synthetic strategy**. Before starting to delineate our findings, it is important to highlight in brief the contributions from many groups to the use of the oxidative-dearomatization of phenols as a powerful tool for the synthesis of nature-inspired scaffolds (Fig. 2a). Among others, Aleman reported an elegant construction of tricyclic fused rings from a cyclohexadienone tethered alkenal by employing an organocatalyzed desymmetrization strategy (Fig. 2a)[26]. Ogoshi described a one-pot desymmetrization of an alkynyl-cyclohexadienone via nickel-catalyzed cyclization in the presence of a chiral ligand[27]. Gaunt and coworkers utilized cyclohexadienone alkanal Michael addition for the asymmetric construction of bicyclohexenones[28].

Inspired by these findings, we herein describe the development of build/couple/pair (B/C/P) approaches that harness the power of three-directional transformations employing metal-free cycloaddition reactions of oxocyclohexa-2,5-dienylpropanal 2a, b, produced by oxidative-dearomatization of the corresponding phenol derivatives (Fig. 3). Thus, we were able to employ four readily accessible building blocks, 2a, 2b, 9 and 11, to assemble a representative library of more than 37 diastereomerically and/or enantiomerically pure members with up to 4 contiguous chiral centers and up to 6 fused rings with a broad distribution of molecular shapes.

**Three-directional B/C/P strategy**. The three-directional route utilized the pluripotent building block II, in which the aldehyde group served as a key starting point which was transformed to the imine nucleofuge III (Fig. 2b). The latter was identified as a suitable branching point for the generation of a set of skeletally diverse scaffolds, IV–VI, each of which contains a functional subunit that sets the stage for a series of subsequent transformations for growing the pilot library. In the first round, a coupling between the imine branch with various amines under acid catalyzed conditions, generated the octahydroindolo[2,3-a]quinolizine IV (Fig. 2b). The latter contains handles useful for differentiation in the post-pairing stage. A second, one-step transformation leading to the diastereoselective preparation of oxocane scaffolds of type V, was discovered. In the third round, employment of amino acid derivatives as one of the reaction components allowed for the rapid access of polysubstituted piperazinones of type VI with complete enantio-control (Fig. 2b, vide supra).

**Synthesis of octahydroindolo[2,3-a]quinolizine architectures**. Borrowing inspiration from our previous reports directed toward privileged substructure diversity-oriented synthesis[29,30], it was envisioned that subjecting the 4-oxocyclohexa-2,5-dienyloxy) acetaldehyde 2a, and variants thereof, to reactions with various tryptamine derivatives should lead to a structurally complex and skeletally diverse octahydroindolo[2,3-a]quinolizine scaffolds (Fig. 3). Thus, hyperiodinate-catalyzed oxidative-dearomatization followed by Dess-Martin oxidation of the phenol precursor 1, produced the cyclohexadione pluripotent building block 2a (Fig. 3). We first examined the reaction of tryptamine (3a) with 2a employing different catalysts (T3P, Sc(OTf)₃), solvents and conditions (−78 to 100 °C), unfortunately, none of these proto-cols indicated the formation of the desired product of type 5 (Fig. 3). Nonetheless, we were delighted to find that using tri-fluoroacetic acid (TFA) as a reaction promoter, rapidly furnished the indolo-fused benzo[b]pyrido[1,2-d][1,4]oxazinone 5a in 67% yield with >99% dr (Fig. 3).

Optimization of the synthetic route was followed; treatment of cyclohexadienone acetoxyl aldehyde 2a with tryptamine (3a) in the presence of 2.0 equiv of TFA at −78 °C provided intermediate 4. Subsequent aza-Michael addition delivered the pentacyclic compound 5a as a single diastereoisomer. The structure of compound 5a was unambiguously confirmed using 1D- and 2D-NMR (Supplementary Figures 16–24) and X-ray crystallographic analysis (Fig. 3 and Supplementary Table 1). The Pictet-Spengler/aza-Michael addition cascade provided a single diastereoisomer, presumably through si-face addition of the β-carboline nitrogen on the enone group in intermediate 4[26] (Supplementary Figure 1).

With the optimized conditions in hand, the 5-methoxy tryptamine (3b) was reacted with the aldehyde 2a, to deliver the pentacyclic compound 5b, as a single diastereoisomer in 69% yield. The reaction was expanded by engaging the isosteric oxygen-tethered aldehyde 2b under a similar reaction with tryptamine to produce a mixture of the diastereoisomers 5c, 6c in a ratio of 7:3 and 73% yield. The structures of which were confirmed through 2D-NMR (Supplementary Figures 27–40) and X-ray crystallographic analysis (Fig. 3 and Supplementary Tables 2 and 3). Similarly, the reaction between 5-methoxy tryptamine and aldehyde 2b yielded the corresponding products 5d and 6d as a mixture of diastereoisomers (Fig. 3) in 7:3 ratio and 73% yield (vide infra). The high diastereoselectivity observed in the case of the carbon-tethered aldehyde 2a compared to the isosteric oxygen-tethered aldehyde 2b, could be rationalized as shown in Supplementary Figure 1. Obviously, in the case of the oxygen-tethered aldehyde 2b, the β-carboline nitrogen atom, approaches the Michael-adduct from the si- or the re-face. This might be attributed to the favored equilibrium between intermediates VIII and XI compared to the disfavored one between intermediates II and V. Clearly, the inherent steric bias in intermediate II, due to the V-shaped methoxyl arm enforces the si-face nucleophilic addition on the α,β-unsaturated system leading to a single diastereoisomer (Supplementary Figure 1).

To enrich the library with compounds bearing a greater number of chiral centers, the asymmetric synthesis of octahy-droindolo[2,3-a]quinolizines was achieved when amino acids were utilized as one of the reaction components (Fig. 4a). Thus, aldehyde 2a was subjected to a reaction with (L)-tryptophan methylester (7a) to produce the corresponding polycyclic octahydroindolo[2,3-a]quinolizine system 8a with four contig-uous chiral centers in 63% yield and >99% ee (Fig. 4a). The structure of 8a was confirmed through extensive 1D- and 2D-NMR analysis (Supplementary Figures 45–53). A similar reaction between the aldehyde 2a and (D)-tryptophan methylester (7b), delivered 8b in 55% yield and > 99% ee. The structure of 8b, was

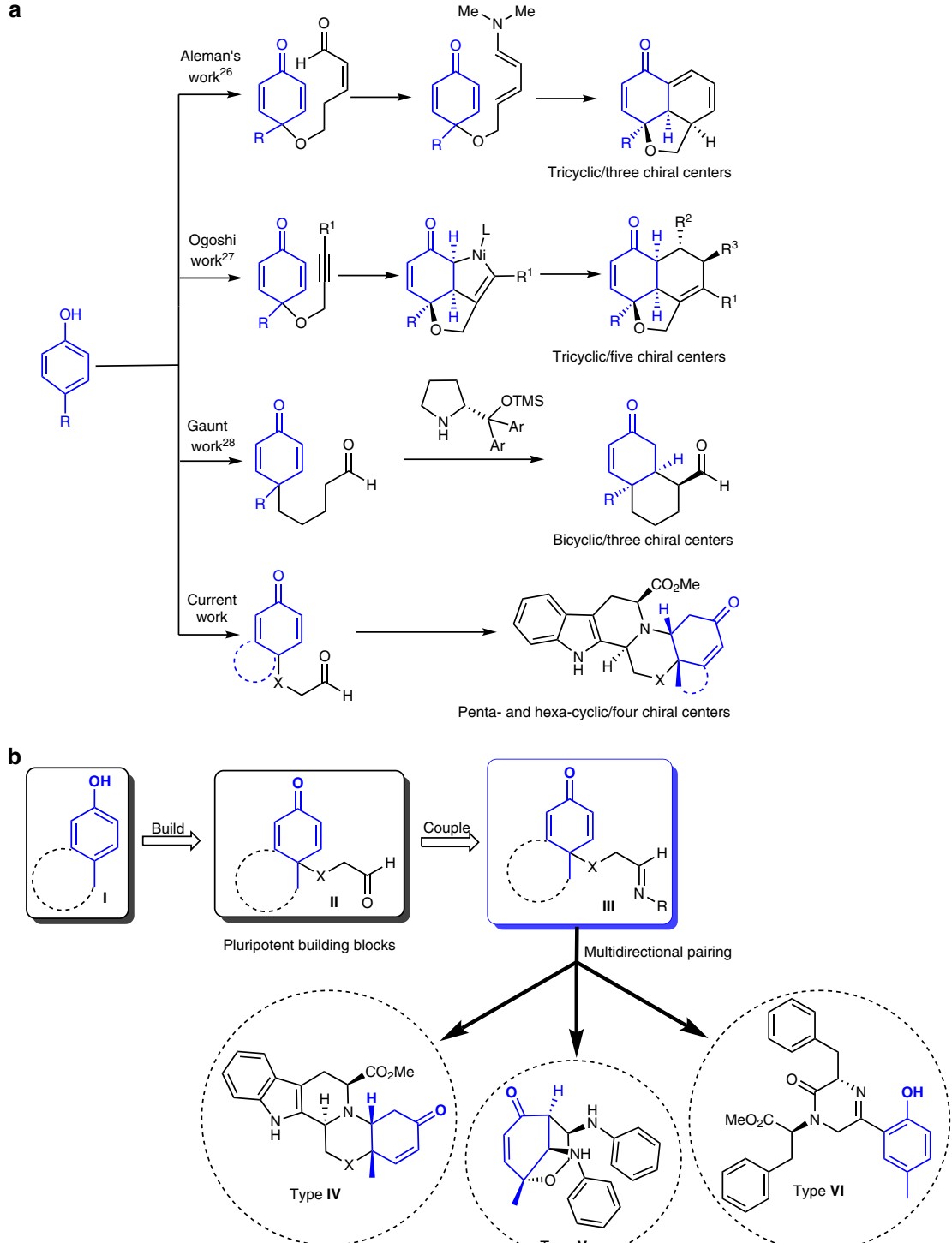

**Fig. 2** Multidirectional utilization of oxidative-dearomatization building blocks. **a** Previous and current approaches utilizing the building blocks derived from oxidative-dearomatization reaction. **b** Three-directional build/couple/pair strategy for the access of diverse molecular shapes

unambiguously confirmed through X-ray crystallographic analysis (Fig. 4a and Supplementary Table 4). The same sequence was also implemented to synthesize the hydroxyl substituted indolo derivative **8c** in 40% yield. To diversify the library with various ring types, the synthesis of enantiomerically pure morpholino-containing analogues, compounds **8d** and **8e** (Fig. 4a) were achieved in 43% and 52% yield and >99% ee, respectively.

To further grow the library with diverse molecular shapes, the hexahydro-7-oxonaphthalen-4a-yloxy)acetaldehyde (**9**) was synthesized and subjected to reaction with tryptamine under the same conditions to enable the synthesis of the hexacyclic indoline system **10** in 55% yield and >99% dr (Fig. 4b). The structure of compound **10** (vide infra) was deduced from its 1D-and 2D-NMR spectra (Supplementary Figures 66–75) as well as

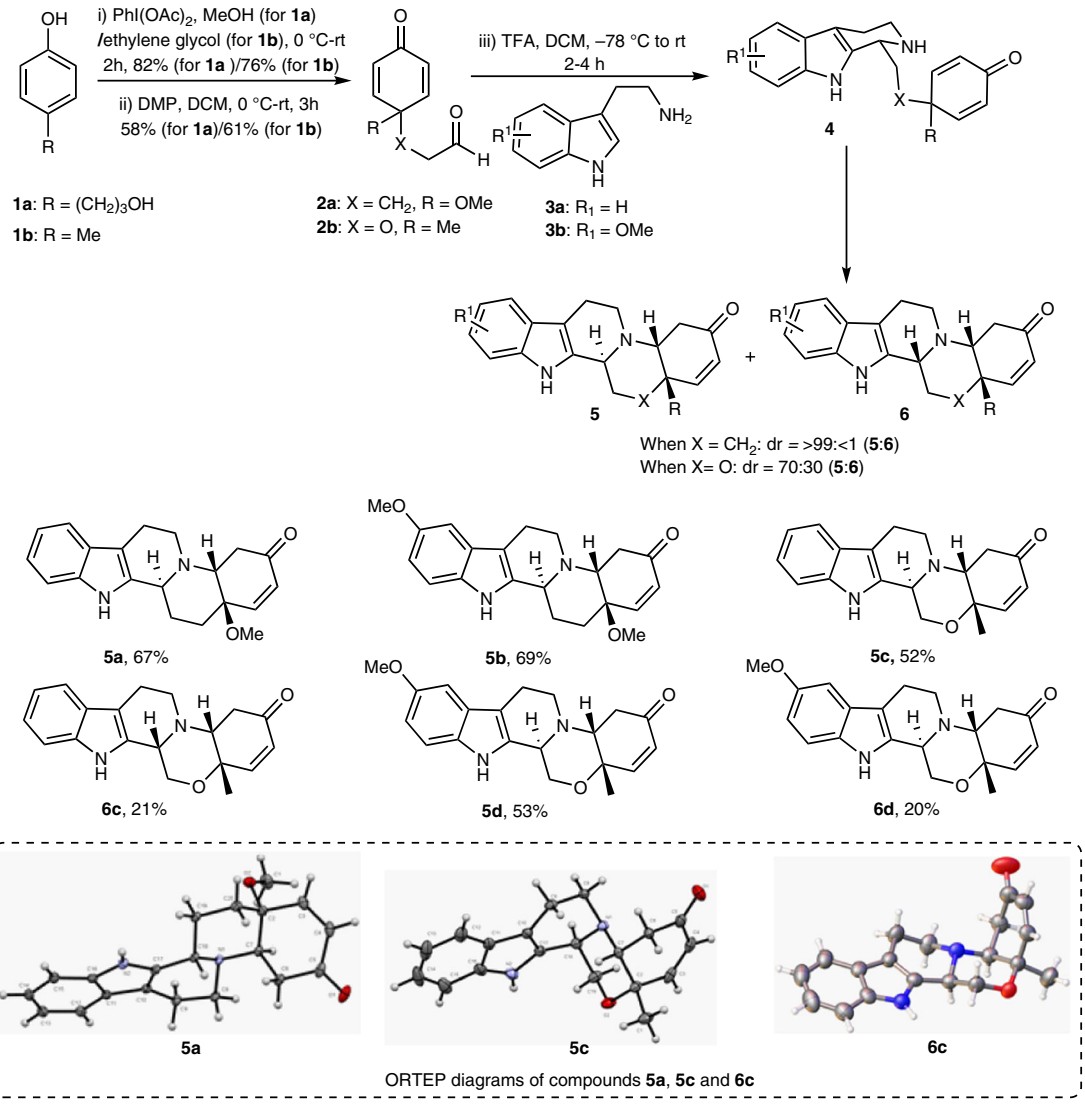

**Fig. 3** Diastereoselective synthesis of polycyclic octahydroindolo[2,3-a]quinolizines. ORTEP diagrams of compounds **5a**, **5c,** and **6c**. PhI(OAc)$_2$, (diacetoxyiodo)benzene; MeOH methanol, DMP Dess-Martin periodinane

X-ray crystallographic analysis (Fig. 4b and Supplementary Table 5).

An interesting discovery was made when the tetrahydro-5-oxo-1H-inden-7a-yloxy)acetaldehyde (**11**) was utilized as a starting material. In this case, a different regioselectivity was observed delivering a 1:1 mixture of diastereoisomers **12** and **13**, in 60% yield (Fig. 4b). The structures of compounds **12** and **13** were unequivocally deduced through X-ray analysis (Fig. 4b and Supplementary Tables 6 and 7). This is notorious to the reaction of compound **9** when reacted with tryptamine (Fig. 4b), in that the aza-Michael addition reaction proceeded at the least hindered site of the α,β-unsaturated producing a single diastereoisomer. These controversial findings could be rationalized as depicted in Supplementary Figure 2. Clearly, the inherent steric bias in both aldehydes **9** and **11**, dictated the reaction regioselectivity. Obviously, the steric compression between the cyclopentane and morpholine rings contained in conformation **IX** is relatively high compared to that present in conformations **XII** and **XV**. This congestion hindered the nucleophilic attack at the least substituted carbon of the α,β-unsaturated system in intermediate **VIII**. However, the minimal steric strain contained in the intermediates **XI** and **XIV** (Supplementary Figure 2), directed

the Michael addition toward the formation of a 1:1 mixture of compounds **12** and **13**, respectively.

**Post-pairing transformations**. The synthesized octahydroindolo[2,3-a]quinolizine pilot library contains a cyclohexenone and β-carboline subunits which provide an opportunity for a subsequent differentiation to enrich the pilot library with diverse molecular shapes. Therefore, our next objective was to identify efficient and robust protocols, which would enable diversification of the library taking advantage of these functionalities, despite the expected variable level of chemical reactivity due to steric bias contained in the octahydroindolo[2,3-a]quinolizine systems. Among various methods examined, two protocols proved to be particularly promising, including the OsO$_4$-mediated and NBS-mediated rearrangement reactions (Figs 5, 6, vide supra). Thus, treatment of compound **5c** with N-methylmorpholine N-oxide (NMO) and a catalytic amount of OsO$_4$ in acetone/water successfully produced the unexpected compound **14** in a quantitative yield as a single diastereoisomer. The structure of the latter was characterized by X-ray crystallography (Fig. 5a and Supplementary Table 8). To the best of our knowledge, such a transformation has never been described before using OsO$_4$ as a catalyst.

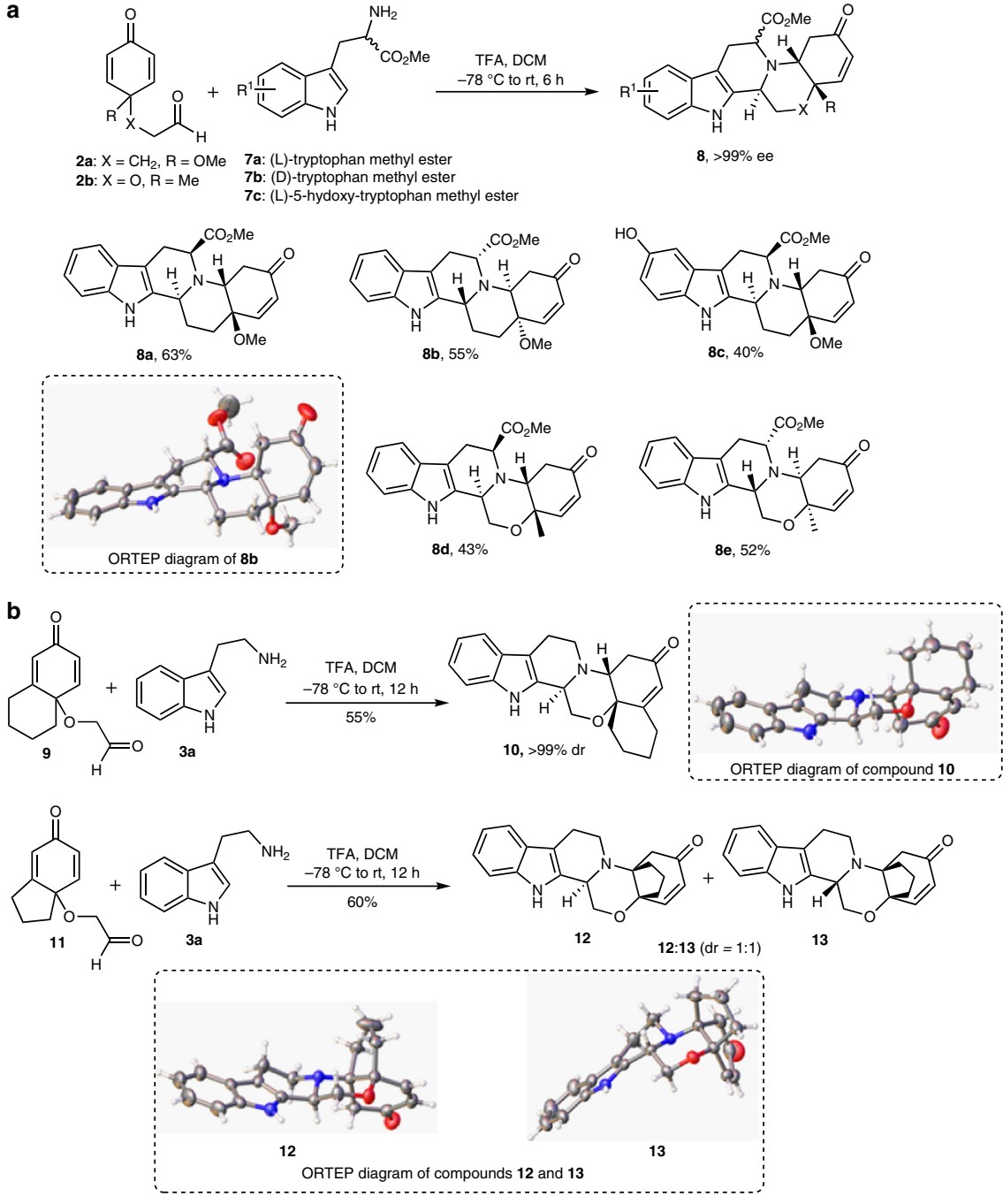

**Fig. 4** Synthesis of skeletally diverse collections of octahydroindolo[2,3-a]quinolizines. **a** Asymmetric synthesis of polycyclic octahydroindolo[2,3-a]quinolizines. **b** Regioselective synthesis of octahydroindolo[2,3-a]quinolizines

Surprisingly, utilizing the same reaction with compound **6c** (Fig. 3), the diastereoisomer of compound **5c**, as a starting material, produced the same stereochemical features contained in compound **14** (Fig. 5a). A plausible mechanism for this transformation is proposed in Fig. 5b. In brief, the *syn*-addition of the two hydroxyl groups was first initiated on the electron-rich pyrrole ring leading to intermediate **II**. Consequently, cascade of events occurred, including protonation, rearrangements, to furnish intermediate **III**. The formation of the latter from the two diastereoisomers, **5c**, **6c**, in which the stereochemistry around the β-carboline ring was lost, explains the delivery of compound **14** from both diastereoisomers. The electron rich intermediate **III**, was then dihydroxylated followed by multiple rearrangements to

deliver intermediate **IV**. Another round of dihydroxylation of the latter followed by multiple rearrangements delivered compound **14**. To expand the scope of this finding, the cycloalkenone **10** was also subjected to the same reaction conditions to deliver the polycyclic compound **15** as a single diastereoisomer in 65% yield.

An attempt was made to carry out the same OsO$_4$ reaction on enantiomerically pure octahydroindolo[2,3-a]quinolizine systems, bearing an ester handle, i.e., compound **8d**, surprisingly, the inherent steric bias imposed by the ester handle directed the reaction toward the conventional *syn*-dihydroxylation function of the OsO$_4$ producing compound **16** in 52% and >99% ee (Fig. 6a, vide infra). The structure of compound **16** was unambiguously concluded from its 1D- and 2D- spectra and X-ray

**Fig. 5** Post-pairing transformations of octahydroindolo[2,3-a]quinolizines. **a** OsO$_4$-catalyzed synthesis of the compounds **14** and **15**. **b** A proposed mechanism for the OsO$_4$ catalyzed formation of compound **14**

crystallographic analysis (Supplementary Figures 110–119 and Supplementary Table 9). To validate this chemoselectivity, compound **8e** was subjected to the same reaction conditions to furnish the corresponding dihydroxylated product **17** as a single enantiomer in 44% yield (Fig. 6a).

To increase the complexity of the library by enriching it with various molecular shapes, a second round of post-pairing transformation was considered. As such, we envisaged the indole subunit contained in these scaffolds to provide a functional handle for NBS-mediated rearrangement reaction to potentially deliver spiro-oxindole scaffolds (e.g., compound **18**)[17]. The latter

represents the basic framework of a wide range of natural products and biologically significant compounds, including anitiviral[31,32], anticancer[33] activities as well as cell cycle inhibitors[34]. For example, spirotryptostatin (Fig. 6b)[32], showed a promising anticancer activity. Thus, treatment of compound **5a** with NBS rapidly rearranged to the desired the pentacyclic spiroxindole **18** as a single diastereoisomer in 51% yield (Fig. 6b)[17]. The structure of the latter was unambiguously confirmed from its 1D-, 2D-NMR spectra (Supplementary Figures 122–130). Encouraged by this complexity-generating reaction accessible primarily via NBS-mediated rearrangement, compound **6c** was subjected to the same

conditions to deliver the morpholino-fused spiro-oxindole system **19** in 52% yield as a single diastereoisomer. The structure of **19** was confirmed by X-ray crystallographic analysis (Fig. 6b and Supplementary Table 10). The same methodology was expanded to access enantiomerically pure spiro-systems. Thus, **8a** was reacted under the same conditions to furnish compound **20** as a single enantiomer in fair yield.

**Synthesis of oxocanes.** The second complexity-generating reaction was discovered when attempting to expand the substrate scope of the cascade. Thus, treatment of **2b** with various anilines (i.e., **21**) carrying electron-donating and electron-withdrawing groups led to the formation of the densely functionalized 8-membered framework, 2-oxabicyclo[3.3.1]non-7-en-6-one **22** (Fig. 7a, vide supra) with four contiguous chiral centers. Thus, a catalyst-free reaction of **2b** with 4-methoxy aniline (**21a**) at -78 °C, surprisingly provided the unexpected oxocane **22a**, in a good yield and complete diastereo-control. The structure of compound **22a** was unambiguously concluded from its 1D-NMR spectra and X-ray crystallographic analysis (Fig. 7a and Supplementary Table 11)). It is worth mentioning that, oxocane architecture exists as the core scaffold in compounds possessing significant biological activities[35,36]. Employing the same conditions to substituted aniline derivatives allowed for the synthesis of compounds **22b**, **22c**, and **22d** in good yields and complete diastereo-control. As a representative example, the relative stereochemistry of compound **22c** was confirmed through extensive 2D-NMR analysis (Supplementary Figures 139–147). An attempt was made to subject the carbon-tethered aldehyde **2a** to the same reaction conditions, unfortunately, a mixture of complex products was observed amid different efforts to optimize the reaction conditions.

A plausible mechanism of this transformation could be delineated as follows: Condensation of the first equivalent of the amine with aldehyde **I** leads to the imine **II**. Aza-Michael addition of the second equivalent of the amine on the cyclohexadienone delivers intermediate **III**. Subsequently, Mannich transannular reaction completed the cyclization reaction to furnish the oxocane **IV** (Fig. 7b).

**Asymmetric synthesis of polysubstituted piperazinones.** An attempt was made to expand the scope of the oxocane generating reaction (Fig. 7a) by using substituted alkyl amines, unfortunately, such reactions proved tedious, amid variations of the

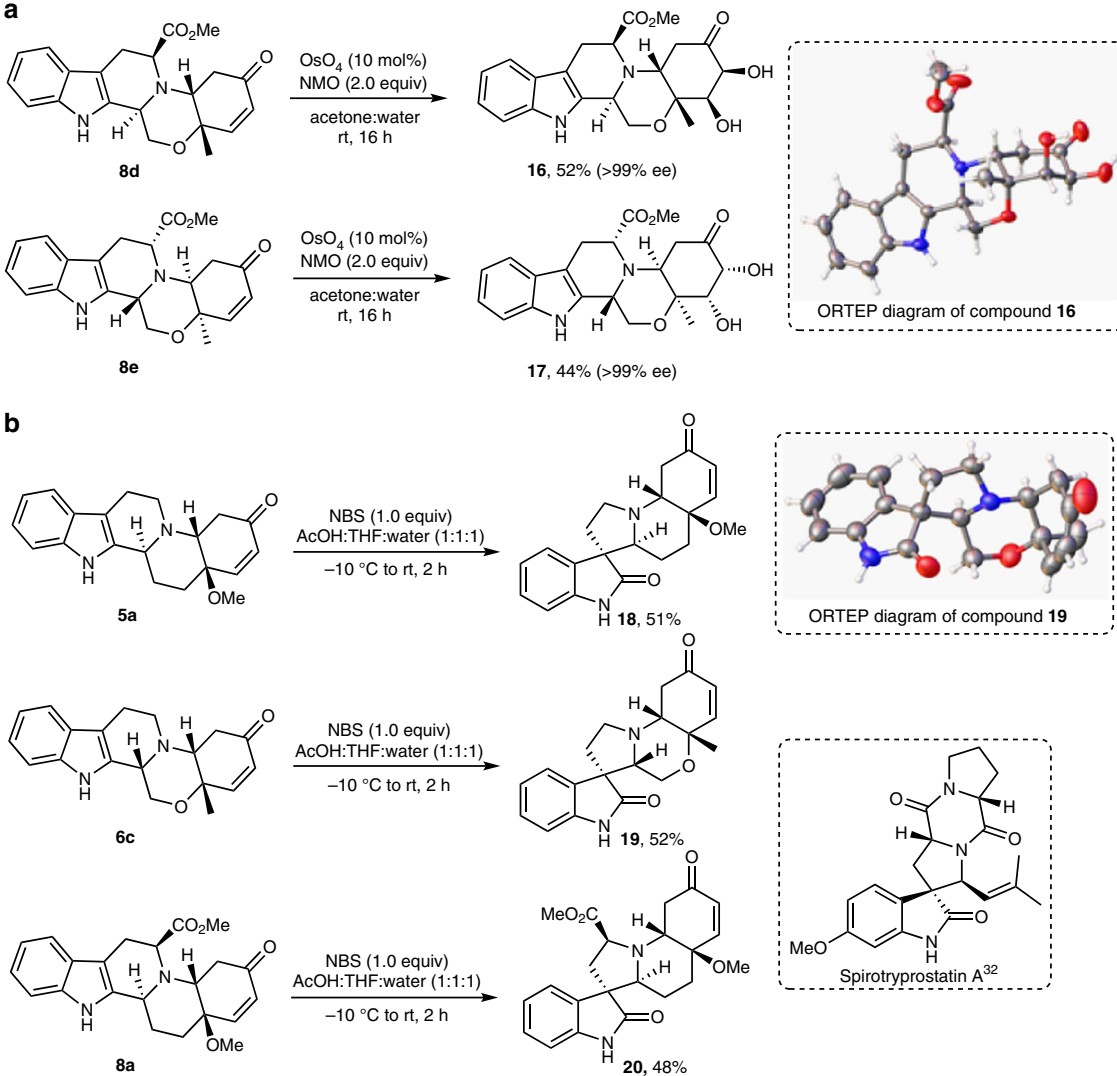

**Fig. 6** Post-pairing transformations of octahydroindolo[2,3-a]quinolizines. **a** OsO₄-catalyzed dihydroxylation. **b** NBS-mediated synthesis of spiro-oxindole polycyclic systems. AcOH acetic acid, THF tetrahydrofuran

**Fig. 7** Tandem synthesis of oxocanes employing aniline and quinone derivatives. **a** Diatereoselective synthesis of oxocanes. **b** A proposed mechanism for the formation of oxocanes. MS molecular sieves

reaction stoichiometry and conditions. We thought this might be due to the strong nucleophilicity of the aliphatic amines compared to their aromatic counterparts. At this junction, we considered the introduction of an ester unit germinal to the amine nucleophile (e.g., esters of amino acids, i.e., **23**), to reduce the nucleophilicity of the amine group, in turn facilitating the construction of the corresponding functionalized oxocanes. Serendipitously, we were delighted to find that using molecular sieves in dichloromethane (DCM) at −78 °C gave a single, yet unexpected product **24a** in 65% yield and in >99% ee (Fig. 8a). Extensive 1D- and 2D-NMR spectroscopic analysis identified this compound to be the piperazinone **24a** (Supplementary Figures 150–158). This serendipitous cascade has turned out to be a general approach to access these complex structures with complete transfer of enantioselectivity and here we report its successful employment using various amino acids and Michael acceptors to generate a set of diversely substituted piperazinones (Fig. 8a, vide infra). Since the piperazinone core exists as the basic nucleus of many important pharmaceutical agents[37], many multistep procedures for their access were reported[38,39]. The described one-step method however, represents an economic, rapid and efficient strategy for the asymmetric synthesis of these scaffolds.

With these findings in hand, we tentatively suggest that the mechanism for the formation of **24** proceeds via a condensation between the amino acid and aldehyde **2b** to produce intermediate **I** (Fig. 8b). Subsequent Claisen rearrangement produced phenol **IV**. Condensation of **IV** with the second equiv. of the amine followed by tautomerization delivered intermediate **VI**. Subsequent transannular cyclization of **VI** through nucleophilic acyl substitution completed the synthesis of the piperazinone scaffold **VII** (Fig. 8b). Several factors led to this mechanistic speculation; principal among them was the fact that this reaction appears to occur slowly with low yield when stoichiometric amount of the

amino acid was used (1.0 equiv). However, a fast and quantitative conversion was observed when 2.0 equiv of the amine was used, indicating that the mechanistic pathway of the cascade is dependent on the amino acid stoichiometry. Furthermore, the enantiomeric purity inherent in the starting material was retained in the final product, providing another concrete evidence of the proposed mechanistic route. At this stage, we are excited to state that, such a cascade, to the best of our knowledge, has not been described before.

Encouraged by this serendipitous complexity-generating reaction accessible primarily via Claisen rearrangement, we decided to employ various amino acid derivatives as one of the reaction components. Thus, we were able to synthesize a pilot collection of six members of this piperazinones family, compounds **24a–f**, containing various substitution patterns in fairly good yields and complete enantio-control (Fig. 8a). To unambiguously confirm the structures of compounds **24a–f**, an X-ray crystallographic analysis (Supplementary Table 12) of compound **24b** was performed and is shown in Fig. 8a, as a representative example.

**Phenotypic screening.** Targeting mitochondrial functions has been implemented as a potential anticancer approach by depriving cancer cells from the energy source[40,41]. Limiting the mitochondrial energy production affects the availability of ATP in cancer cells and consequently the biosynthesis of macromolecules required for tumor growth and survival[42,43]. While several approaches to identify modulators of mitochondrial functions have been developed, researchers mostly relied on screening of known compounds[44,45]. The approach presented here however, aims to evaluate and develop first-in-class chemical probes and to study their effects on mitochondrial functions to target cancer cells proliferation and survival.

In this respect, we examined the ability of the octahydroindolo [2,3-a]quinolizine small molecule library to enable the

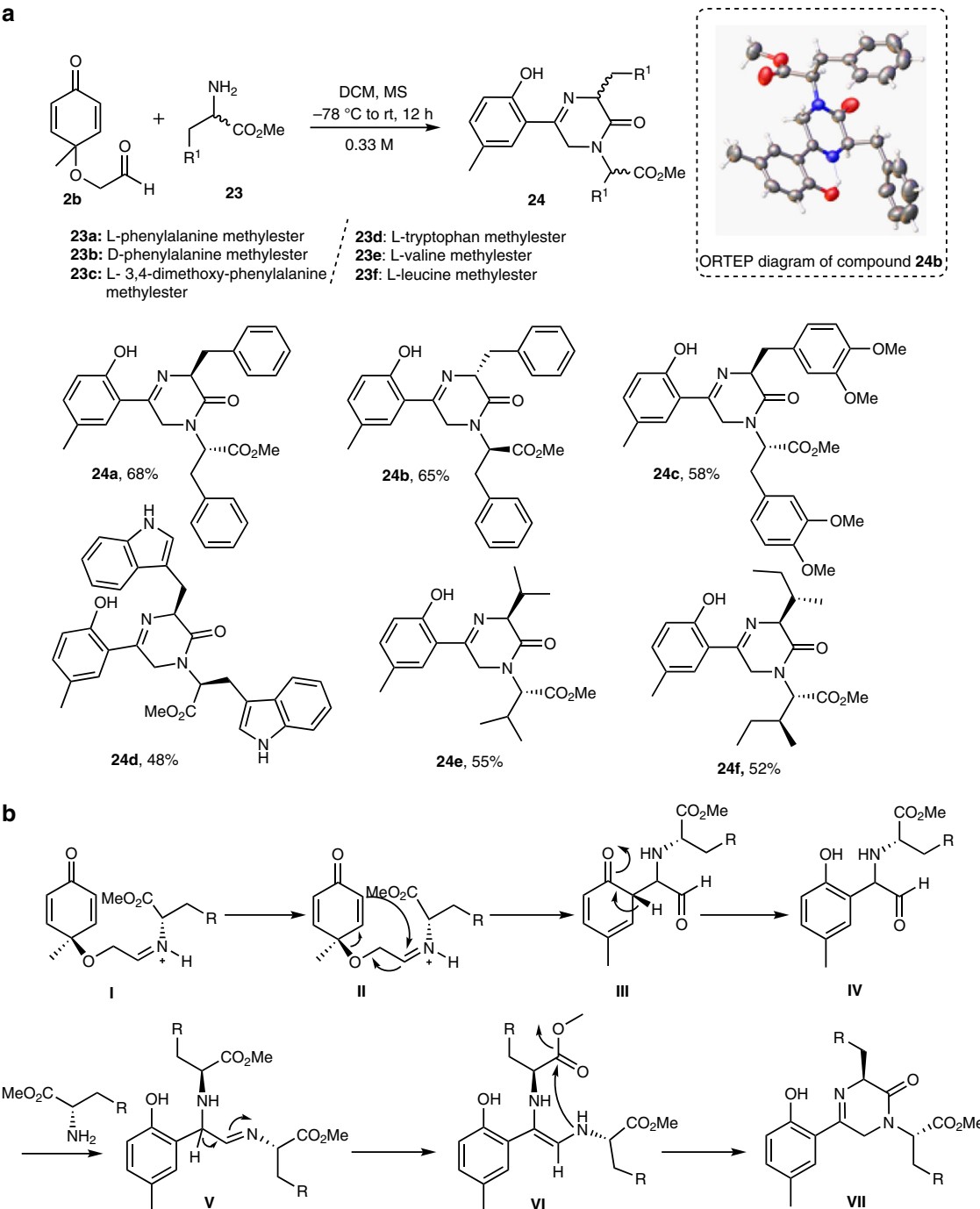

**Fig. 8** Cascade transformation of quinones to piperazinones utilizing amino acids as substrates. **a** Asymmetric synthesis of piperazinones. **b** A proposed mechanism for the formation of piperazinones. MS molecular sieves

identification of chemotypes as potential inhibitors of mitochondrial functions. Such compounds would be useful not only as chemical probes targeting cellular energy metabolism but also as potential leads for the development of drugs targeting the mitochondrial function in cancer. Thus, the cellular mitochondrial activity was measured in Hepa1–6 cell line (CRL-1830, ATCC) after exposure to the octahydroindolo[2,3-a]quinolizine pilot library for 24 h. These efforts identified compounds **5c** and **26c** (Supplementary Figure 3b) as promising leads, which were found to deplete ATP production up to 60% and 50% (Fig. 9a), and reduce $\Delta\Psi$m by 82% and 77%, respectively (Fig. 9b). The

observed increase in redox potential (Fig. 9c), as a result of oxidative stress and accumulation of reducing equivalents, supports the idea of suppression and cellular stress response induction in Hepa1–6 cells due to the effect of compounds **5c** and **26c**. Additionally, the exposure of Hepa1–6 cells to **5c** and **26c** for 48 h resulted in a potent suppression of cells proliferation (Fig. 9d) without noticeable cytotoxic necrosis (Fig. 9e). Furthermore, the octahydroindolo[2,3-a]quinolizine compounds **5c** and **26c** were shown to diminish the proliferation of Hepa1–6 to different degrees with compound **5c** being the most potent.

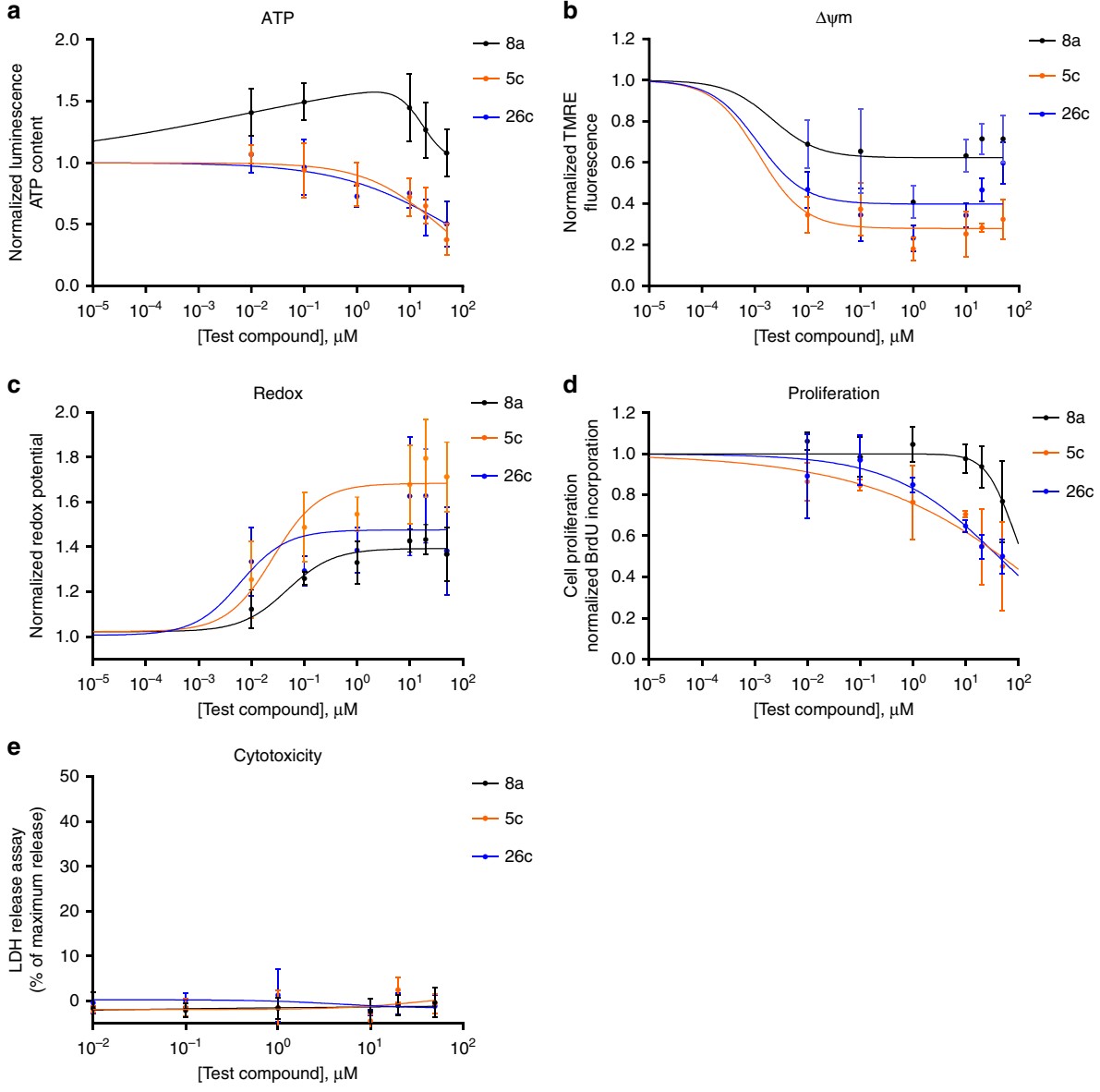

**Fig. 9** Screening of a pilot library on cellular functions in Hepa1–6. **a** Effect of compounds **5c** and **26c** on ATP. **b** Effect of compounds **5c** and **26c** on mitochondrial membrane potential. **c** Effect of compounds **5c** and **26c** on cellular redox potential. **d** Effect of compounds **5c** and **26c** on cellular proliferation. **e** Cytotoxicity of compounds **5c** and **26c**. Error bars indicate standard deviation based on three replicated calculations. Significance was tested using an ANOVA test, with Dunnett's multiple comparison test

Since ATP is the main product of the mitochondria as the cells powerhouse[46], the decrease in ATP content indicated the ability of compounds **5c** and **26c** to interfere with the machinery of cellular energy metabolism. This was validated by a parallel inhibition of the ΔΨm with the maximum response observed at a concentration of 1 μM decreasing the normalized ΔΨm in the test system by 90% (**26c**), 82% (**5c**) (Fig. 9b). In addition, **5c** and **26c** strongly affected the redox potential to a point where the cells succumb to the induced stress (Fig. 9c). The inhibition of glycolysis and the consequent downregulation of ATP are utilized not only for targeting cancer cells but also to sensitize the resistant cancer cells to classical chemotherapies such as, doxorubicin, cisplatin, and paclitaxel.

Cancer immunotherapy is currently one of the most promising and growing approaches in cancer treatment. However, resistance to immunotherapy represents a major concern towards the

maximal patient benefits[47]. The dysregulated metabolism in cancer cells has been linked to immunotherapy resistance[40,48]. The use of classical adjuvant chemotherapeutic agents to sensitize cancer cells to immunotherapy is limited due to immunosuppressant effects. In addition, it has been recognized that mitochondrial metabolism is critical in the regulation of immune responses[49]. Therefore, after showing the ability of compounds **5c** and **26c** to target mitochondrial machinery, we tested their effect on T- and B-cells activation and proliferation. Interestingly, compounds **5c** and **26c** did not suppress T- and B-cell function as indicated by the non-significant effects on the division indices (Fig. 10a, c) or expansion indices (Fig. 10b, d) of T- and B-cells upon stimulation using phorbol 12-myristate 13-acetate (PMA) (PMA)/Ionomycin. In addition, treatment of isolated T-cells with **5c** and **26c** did not significantly affect the activation-induced proliferation upon stimulation of T-cells with either anti-CD3/

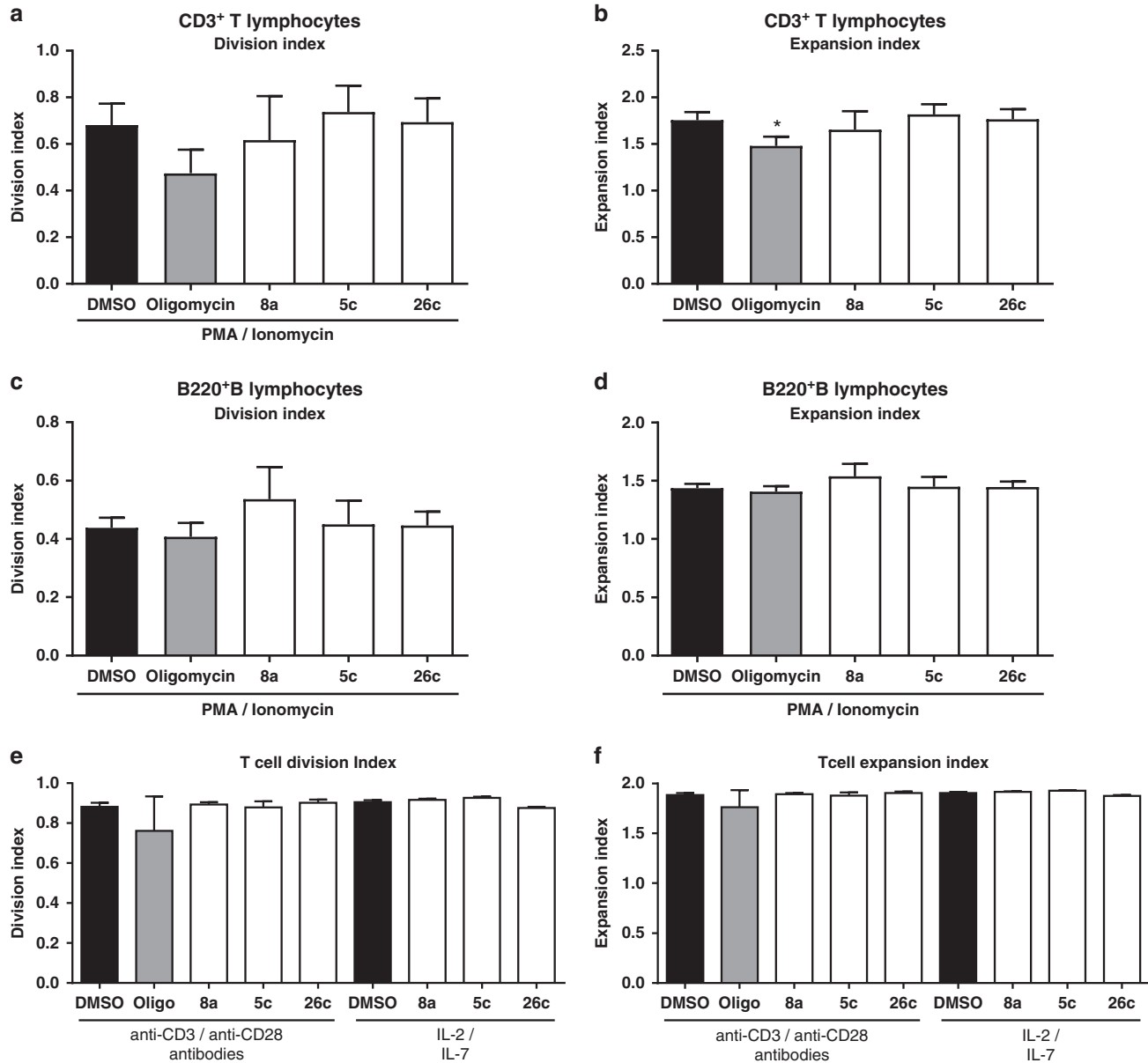

**Fig. 10** Screening of a pilot library on cellular functions in Hepa1–6. **a** Effect of compounds **5c** and **26c** on immune cell proliferation in lymphocytes or isolated T cells. **a**, **b** Effect of compounds **5c** and **26c** on CD3$^+$ T lymphocytes. **c**, **d** Effect of compounds **5c** and **26c** on B220$^+$ B lymphocytes. **e**, **f** Effect of compounds **5c** and **26c** on isolated T cells. Error bars indicate standard deviation based on three replicated calculations. Significance was tested using an ANOVA test, with Dunnett's multiple comparison test. *$p < 0.05$

CD28 antibodies or cytokines IL-2/IL-7 (Fig. 10e, f). Thus, while retaining the suppressive effect on the proliferation of the highly metabolically active Hepa1–6, these compounds did not suppress T- and B-cell activation. Thus, the overall impact of **5c** and **26c** on metabolism appears to be a driving force in the inhibition of cell proliferation[50]. This inhibition of mitochondrial functions in cancer cells without altering the immunological activation or reprogramming of T- and B-cells, represents a promising strategy in cancer immunotherapy. While an in-depth evaluation of these substances is required, the presented findings support a promising application of these compounds as lead drug candidates for the treatment of cancer as well as autoimmune and chronic inflammatory diseases.

In summary, the work presented in this article serves to illustrate the potential power of three-directional synthesis in DOS. The use of this approach allowed us to access a range of

mono-, tri- and penta-cyclic molecular architectures, rapidly and efficiently, following a starting material-based reaction schemes. This concept was validated by a production of a 37-membered library with a broad distribution of molecular shapes starting from only four readily accessible building blocks. The effective introduction of reagent-based diversification into the strategy was extremely satisfying, as by altering the choice of starting material and reaction conditions, we were able to adjust the chemo-selectivity and stereo-selectivity of the reactions to generate a library of a large distribution of molecular shapes. This combination of reagent and condition-based approaches for the generation of molecular diversity afforded many interesting possibilities not achievable by either approach alone. Subsequent cellular screening of the octahydroindolo[2,3-a]quinolizine pilot library identified unique chemotypes, compounds **5c** and **26c** that effectively suppressed glycolytic production of ATP and

membrane potential in Hepa1–6 representing promising lead drug candidates for cancer treatment. Finally, the chemistry described here is expected to facilitate the discovery of chemotypes and work along these lines is currently underway in our laboratories.

## Methods

**General**. Chemical reagents and anhydrous solvents were purchased from Sigma-Aldrich and were used without further purification. Solvents for extraction and column chromatography were distilled prior to use. TLC analysis was performed with silica gel plates (0.25 mm, E. Merck, 60 $F_{254}$) using iodine and a UV lamp for visualization. Retention factor ($R_f$) values were measured using a $5 \times 2$ cm TLC plate in a developing chamber containing the solvent system described. Melting points were measured with a Stuart Melting Point Apparatus (SMP30) in Celsius degrees and were uncorrected. $^1$H, $^{13}$C NMR, and 2D-NMR experiments were performed on a 500 MHz instrument. Chemical shifts are reported in parts per million (ppm) downstream from the internal tetramethylsilane standard. Spin multiplicities are described as s (singlet), d (doublet), dd (double doublets), t (triplet), (td) triple doublets or m (multiplet). Coupling constants are reported in Hertz (Hz). ESI mass spectrometry was performed on a Q-TOF high-resolution mass spectrometer or Q-TOF Ultim LC-MS. Optical rotations were measured with a digital polarimeter using a 100 mm cell of 10 mL capacity. Single-crystal X-ray diffraction data were collected using an Oxford Diffraction XCalibur, equipped with (Mo) X-ray Source ($\lambda = 0.71073$ Å) at 293(2) K.

## Data availability

The X-ray crystallographic coordinates for structures reported in this article have been deposited at the Cambridge Crystallographic Data Centre (CCDC), under deposition number CCDC 1857621 for **5a**, CCDC 1857622 for **5c**, CCDC 1857624 for **6c**, CCDC 1857606 for **8b**, CCDC 1857719 for **10**, CCDC 1857718 for **12**, CCDC 1857721 for **13**, CCDC 1857730 for **14**, CCDC 1857748 for **16**, CCDC 1857869 for **19**, CCDC 1857753 for compound **22a** and CCDC 1857755 for **24b**. These data can be obtained free of charge from The CCDC via (https://www.ccdc.cam.ac.uk/data_request/cif). The authors declare that other data supporting the findings of this study are available within the paper and its supplementary information files and also are available from the corresponding author upon request.

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

## Acknowledgements

This work was supported by generous grants from the Research Funding Department, University of Sharjah, UAE (15011101007 and 15011101002).

## Author contributions

T.H.A designed the synthetic strategies. T.H.A and S.I. designed the biological research with assistance from V.S., P.S., and H.O. P.S., S.I., and H.O. analyzed the biological data. Synthetic chemistry was performed by V.S. and A.S. T.H.A and S.M.S assisted in proposing the mechanisms, stereochemical analysis, and article writing. Experimental X-ray and its analysis was done by M.A.K and R.A.A. M.J.O. carried out HRMS experimental and interpretation. T.H.A. wrote the manuscript with contributions from all authors. All authors were actively engaged in editing the manuscript and gave their approval of the final version.

## Additional information

**Competing interests:** The authors declare no competing interests.

