## [Peer Review File · Nature Communications]

Reviewer #1 (Remarks to the Author):

The manuscripts reports the structures of 12 crystal structures to support the reported chemistry. Unfortunately all structures use generic computer generated atom labels which makes reviewing more difficult, however in general the structures establish atom connectivity and relative configuration of stereocentres (these are light atom structures and the radiation source is Molybdenum which means that differentiating between enantiomers is difficult). There are a few structures for which the data completeness is well below the IUCR requirements of 98.5% (some as low as 76% complete), while this does not mean the structures are wrong it does suggest a bit of sloppiness on behalf of the crystallographer. Structure with CCDC code 1857718 (Compound 12) has been reported in the triclinic space group P-1, with two molecules in the asymmetric unit, this is incorrect, and should be in the space group P21/c with one molecule in the asymmetric unit. I have attached a shelx .res file with the transformation matrix to convert the data to this space group. This structure must be refined in this space group and the resulting cif uploaded to replace the existing one.

The cif provided for compound 24a (1857755) has the opposite configuration to that indicated in Figure 5b, and is also opposite to the ortep inset given for 24a in the paper.

For compound 8a, the ortep diagram given in the paper, and the cif (1857606) provided are of the opposite configuration to that given in the chemical structure diagram for 8a

These issues must be attended to and clarified, particularly for 8a and 24a for which the structures provided actually correspond to 8b and 24b respectively.

Structure (1857869) has a long aromatic C-H bond of 1.37 Å, it appears that all the H atoms have been allowed to refine freely which is OK usually, but the data for this crystal is low quality and as a result one of the H atoms has drifted. All H atoms attached to carbon should be fixed into idealised positions, the resulting cif should then be uploaded to CCDC to replace the current cif which has CCDC code 1857869.

Reviewer #2 (Remarks to the Author):

The manuscript by Al-Tel and coworkers demonstrates the power of synthetic planning to produce a versatile template that can be transformed into structurally diverse and complex products that are comparable to certain nitrogen-containing natural products.

Overall this is a nice illustration of DOS and the products generated are of a highly structurally complex nature-perhaps of a structural complexity unseen by previous efforts. The diastereoselectivities of the bond forming reactions are high. Furthermore, the structural assignments are well supported by X-ray crystallographic studies. That said, it is this reviewer's view that the underlying concept of this work is not novel. Schreiber's review on the build/couple/pair approach (B/C/P) to DOS is now over a decade old. While this work revolves around some elegant synthetic transformations it does not conceptually go into a new direction from the B/C/P approach. Moreover, several of the transformations were unpredicted making the work less general. Therefore, it is not believed that this work will find broad appeal to the chemistry or chemical biological community.

The manuscript is well written and but in light of the above discussion, this is potentially better served as a full article (perhaps in JACS). Given the narrow scope and lower novelty of work this reviewer does not feel that the work will be considered of interest to the larger synthetic chemistry or chemical biology community. As such it is my recommendation that this work not be accepted for publication in Nature Communications given that it is more suited to a specialized journal.

Reviewer #3 (Remarks to the Author):

The step-economical synthesis of complex and sp³-rich natural product-inspired libraries is a challenge to the synthetic organic chemists. Moreover, one step Diversity Oriented Synthesis (DOS) from privileged scaffolds or an intermediate to diverse molecular structures is an art and an important milestone for drug design and discovery. Symmetrical cyclohexadienones, synthesized via oxidative-dearomatization of phenols, is such as privileged structure employed previously for polycyclic compounds using transition metals and organocatalysis.

In the present manuscript by Al-Tel et al, entitled "Multidirectional Desymmetrization of Pluripotent Building Block en Route to the Asymmetric/Diastereoselective Synthesis of Complex Nature-Inspired Scaffolds" described one-step access to diversely functionalized octahydroindolo[2,3-a]quinolizine scaffolds in high diastereoselectivity via desymmetrization of symmetrical oxocyclohexa-2,5-dienylpropanal (2) with tryptamine following the cascade process of Pictet-Spengler and aza-Michael cyclizations. Chiral amines, tryptophane was also employed for the synthesis of their enantio-enriched variants with complete transfer of chirality (99% ee). Unsymmetrical cyclohexadienones (9 and 11) were employed to achieve bridged analogues with excellent diastereoselectivity. These complex penta/hexacyclic compounds were achieved in low to moderate chemical yields on average, however, in very good diastereoselectivity. The several key (viz. OsO₄ and NBS mediated) organic

transformation were performed on octahydroindolo[2,3-a]quinolizine scaffolds to archive rearranged products, such as novel amino-acetals and spiro-polycyclic compounds, respectively. In addition, these privileged cyclohexadienone-aldehydes were serendipitously enabled to access the synthesis of oxocanes and piperazinones with the variation of amine coupling partners, probably due to remarkable reactivity difference of aromatic and aliphatic amines.

Overall, the developed strategy demonstrated the synthesis of diverse range of scaffolds (37 products), starting from a privileged cyclohexadienone-aldehydes (4 building blocks) and amines. The reaction conditions are simple and diverse molecular scaffolds were archived by varying the structure of amines. Under acidic condition, amino acids gave respective octahydroindolo[2,3-a]quinolizine scaffolds, whereas in the absence of acid, piperazinones were formed. In contrast, aromatic amines led to oxocanes. Although, chemical yields are not much appealing in most of examples, excellent diastereoselectivity, diversification of scaffolds, and simple reaction conditions are quite impressive. Finally, phenotypic screening of the was done with octahydroindolo[2,3-a]quinolizine library and authors have claimed that compounds 5c and 26c that effectively suppressed glycolytic production of ATP and membrane potential in hepatoma cell line. I didn't see any structure corresponding to 26c ??

The chemistry described here is interesting and quite appealing to the synthetic as well as medicinal chemists for design and synthesis various sp³ rich molecular scaffolds. Thus, in my opinion, the present manuscript can be accepted after following addition and corrections.

1) Abstract, line 12: "amino acids" were used as chiral reagent, not auxiliaries, needed to be changed.

2) Scheme 3a; Yields for 2a and 2b should be clearly mentioned.

3) Scheme 3c; Isn't the aza-Michael reactions of intermediate, derived from 9 and 11 shows "regioselectivity", not "chemoselectivity" ?? Mechanism part for the explanation of switching of regioselectivity with cyclopentane and cyclohexane analogues is not so clear. The intermediate XII shows some steric congestion b/w 'H' and cyclopentane-bridged methylene groups??

4) Page 13; It is mentioned NBS-catalysed, however it is not. It should be changed to NBS-mediated.

5) It is mentioned that ". This serendipitous cascade has turned out to be a general approach to access these complex structures with complete enantioselectivity.....". However, it would be more appropriate to mention "...with complete transfer of enantioselectivity".

6) From the library of octahydroindolo[2,3-a]quinolizine, 5c and 26c are most effective to suppress glycolytic production of ATP and membrane potential in hepatoma cell line, however structure 26c is not shown ?? Special attention is needed to denote the number for hit molecule. The presentation of tabular data of activity of other analogues will be more informative for medicinal chemists, it should be mentioned.

7) Synthetic procedures and spectral data should be presented for the starting materials (2a, 2b, 9, and 11) in Supplementary Information.

Responses to Reviewer 1 comments:

Comment: Structure with CCDC code 1857718 (Compound 12) has been reported in the triclinic space group P-1, with two molecules in the asymmetric unit, this is incorrect, and should be in the space group P21/c with one molecule in the asymmetric unit. I have attached a shelx .res file with the transformation matrix to convert the data to this space group. This structure must be refined in this space group and the resulting cif uploaded to replace the existing one.

Response: Thank you very much for the positive and supportive comments. We have refined the structure as requested to be in the space group P21/c with one molecule in the asymmetric unit. Furthermore, the original cif file of compound **12** that was deposited in the CCDC is now replaced by the refined cif file for the CCDC 1857718. During the refining process some parameters values were changed and these new values were reported in the corresponding table of compound **12** in the SI (Supplementary Table 6, Page S17-S18). Also the new ORTEP is now included in Figure 3c (Revised manuscript, Page 11).

Comment: The cif provided for compound **24a** (1857755) has the opposite configuration to that indicated in Figure 5b, and is also opposite to the ortep inset given for **24a** in the paper. For compound **8a**, the ortep diagram given in the paper, and the cif (1857606) provided are of the opposite configuration to that given in the chemical structure diagram for **8a**. These issues must be attended to and clarified, particularly for **8a** and **24a** for which the structures provided actually correspond to **8b** and **24b** respectively.

Response: Thank you very much for the helpful comments. After revising the original X-ray data in the X-ray machine, we found that the data we reported for compounds **8a** and **24a**, in the original submission were right however, they belong to compounds **8b** and **24b**. We mistakenly assigned them to **8a** and **24a**. Please accept our apology for such a mistake and we would like to thank the reviewer for this important note. Now both typos were corrected in the text (page 12, paragraph 2; page 21, paragraph 2). The ORTEP drawings were now assigned to compounds **8b** (Fig. 3b) and **24b** (Fig. 5c).

Comment: Structure (1857869) has a long aromatic C-H bond of 1.37 Å, it appears that all the H atoms have been allowed to refine freely which is OK usually, but the data for this crystal is low quality and as a result one of the H atoms has drifted. All H atoms attached to carbon should be fixed into idealised positions, the resulting cif should then be uploaded to CCDC to replace the current cif which has CCDC code 1857869.

Response: Thank you very much for this supportive comment. As suggested, the X-ray structure of compound **19** was refined and the drifted aromatic C-H bond was refined. The refined cif file was uploaded in the CCDC and the old cif file was replaced and the new refined file number is CCDC 1857869. During the refinement process, some parameter values were changed and the new values were placed in the Supplementary Table 10 for compound **19**. Furthermore, the new ORTEP is placed in Fig. 4d.

Responses to Reviewer 2 comments:

Comment: The manuscript by Al-Tel and coworkers demonstrates the power of synthetic planning to produce a versatile template that can be transformed into structurally diverse and complex products that are comparable to certain nitrogen-containing natural products.

Overall this is a nice illustration of DOS and the products generated are of a highly structurally complex nature-perhaps of a structural complexity unseen by previous efforts. The diastereoselectivities of the bond forming reactions are high. Furthermore, the structural assignments are well supported by X-ray crystallographic studies. That said, it is this reviewer's view that the underlying concept of this work is not novel. Schreiber's review on the build/couple/pair approach (B/C/P) to DOS is now over a decade old. While this work revolves around some elegant synthetic transformations it does not conceptually go into a new direction from the B/C/P approach. Moreover, several of the transformations were unpredicted making the work less general. Therefore, it is not believed that this work will find broad appeal to the chemistry or chemical biological community.

The manuscript is well written and but in light of the above discussion, this is potentially better served as a full article (perhaps in JACS). Given the narrow scope and lower novelty of work this reviewer does not feel that the work will be considered of interest to the larger synthetic chemistry or chemical biology community. As such it is my recommendation that this work not be accepted for publication in Nature Communications given that it is more suited to a specialized journal.

Response: We thank the reviewer very much for the positive and supportive comments.

Responses to Reviewer 3 comments:

Comment: Finally, phenotypic screening of the was done with octahydroindolo[2,3-a]quinolizine library and authors have claimed that compounds 5c and 26c that effectively suppressed glycolytic production of ATP and membrane potential in hepatoma cell line. I didn't see any structure corresponding to 26c ??

Response: We would like to thank the reviewer for the very supportive, positive and valuable comments. Compound **26c** is present in the Supplementary Fig. 3b (page S33, Revised Supplementary Information). Furthermore, a sentence was included for the same in the revised manuscript page 22 line 14-15.

Comment: Abstract, line 12: "amino acids" were used as chiral reagent, not auxiliaries, needed to be changed.

Response: Thank you. As suggested, the term "auxiliaries" is now changed to read "reagents" in the revised manuscript (Abstract, page 2, line 12).

Comment: Scheme 3a; Yields for **2a** and **2b** should be clearly mentioned.

Response: Thank you very much for the note. As suggested, yields for compound **2a** and **2b** are now included in Figure 3a.

Comment: Scheme 3c; Isn't the aza-Michael reactions of intermediate, derived from 9 and 11 shows "regioselectivity", not "chemoselectivity" ??

Response: Thank you for this valuable comment. As suggested, the term "chemoselectivity" is corrected to read "regioselectivity" in the legend of the Fig. 3c and also in the revised manuscript page 13, line 4; page 13, line 10.

Comment: Mechanism part for the explanation of switching of regioselectivity with cyclopentane and cyclohexane analogues is not so clear. The intermediate XII shows some steric congestion b/w 'H' and cyclopentane-bridged methylene groups??

Response: Thank you very much for this helpful comment. We have now revised the part related to the regiochemistry of compounds **10** and **11** (page 13, 2nd paragraph). The revised text now reads as "Obviously, the steric compression between the cyclopentane and morpholine rings contained in

conformation **IX** is relatively high compared to that present in conformations **XII** and **XV**. This congestion hindered the nucleophilic attack at the least substituted carbon of the α,β -unsaturated system in intermediate **VIII**. However, the minimal steric strain contained in the intermediates **XI** and **XIV** (Supplementary Figure 2, S32), directed the Michael addition toward the formation of a 1:1 mixture of compounds **12** and **13**, respectively.”

Comment: Page 13; It is mentioned NBS-catalysed, however it is not. It should be changed to NBS-mediated.

Response: Thank you. As suggested, the term “NBS-catalyzed” is now corrected in the revised manuscript to read as “NBS-mediated” in page 13, paragraph 2.

Comment: It is mentioned that “. This serendipitous cascade has turned out to be a general approach to access these complex structures with complete enantioselectivity.....”. However, it would be more appropriate to mention “...with complete transfer of enantioselectivity”.

Response: Thank you very much. As suggested, the sentence is now modified in the revised manuscript to read as “This serendipitous cascade has turned out to be a general approach to access these complex structures with complete transfer of enantioselectivity and here we report its successful employment using various amino acids and Michael acceptors to generate a set of diversely substituted piperazinones (Fig. 5c, *vide infra*)” (please see page 20, line 15).

Comment: From the library of octahydroindolo[2,3-a]quinolizine, 5c and 26c are most effective to suppress glycolytic production of ATP and membrane potential in hepatoma cell line, however structure **26c** is not shown ?? Special attention is needed to denote the number for hit molecule. The presentation of tabular data of activity of other analogues will be more informative for medicinal chemists, it should be mentioned.

Response: Thank you very much for the comment. The structure of compound **26c** is now shown in Fig. 3b in the supplementary information page 33 and this was also stated in the revised manuscript page 22, lines 8-9. Furthermore, as suggested, the activities parameters for the analogs are now presented in the revised supplementary information (Supplementary Table 13, page S31).

Comment: Synthetic procedures and spectral data should be presented for the starting materials (**2a**, **2b**, **9**, and **11**) in Supplementary Information.

Response: Thank you very much for this comment. As suggested, detailed schemes (Supplementary Schemes 1 to 4), experimental procedures, full characterization data and spectral copies for the starting materials **2a**, **2b**, **9** and **11** are now included in the revised supporting information (pages S34-S35; S41-S42; S43-S44). The Spectra of compounds **2a**, **2b**, **9** and **11** are also included in the supporting information (pages S60-S63, S89-S90 and S97-S98).

Reviewer #1 (Remarks to the Author):

X-ray structures determinations are now acceptable

Reviewer #3 (Remarks to the Author):

All of my comments are well addressed and included in the manuscript.

Please include some following corrections in the revised format before publication

- i) figure 3a in SI as well as in the manuscript should be changed to "Scheme 3"
- ii) Some of structures in SI are not clearly visible (atoms, O and H) ; eg. 1c, 9b, and 11b etc (page 60, 63, 89,. 97 etc)
- iii) ^1H and ^{13}C NMR for compounds 2a contain some aromatic impurities (S62). Pure spectra should be placed in SI.

Ravindra

a

Responses to the Reviewers comments:

Reviewer #1

Comment: X-ray structures determinations are now acceptable

Response: We thank the reviewer for the positive support and for accepting the modified Xray structural data.

Reviewer #3

All of my comments are well addressed and included in the manuscript. Please include some following corrections in the revised format before publication

Comment: figure 3a in SI as well as in the manuscript should be changed to "Scheme 3"

Response: We thank the reviewer for this comment. As such, we followed the journal style as stated in the editor's comments. It's not allowed to use "Scheme", neither in the manuscript text nor in the supplementary information.

Comment: Some of structures in SI are not clearly visible (atoms, O and H) ; eg. 1c, 9b, and 11b etc (page 60, 63, 89,. 97 etc)

Response: We thank the reviewer for this comment. As suggested, all the structures in SI are redrawn to be clearly visible according to the Nature journal template.

Comment: ^1H and ^{13}C NMR for compounds **2a** contain some aromatic impurities (S62). Pure spectra should be placed in the Supplementary.

Response: We thank the reviewer for this comment. As required, clean ^1H and ^{13}C NMR spectra of compound **2a** has been provided in the Supplementary Information (Supplementary Figures 10 and 11).